# Increased Overall Mortality Even after Risk Reducing Surgery for *BRCA*-Positive Women in Western Sweden

**DOI:** 10.3390/genes10121046

**Published:** 2019-12-16

**Authors:** Anna Öfverholm, Zakaria Einbeigi, Antonia Wigermo, Erik Holmberg, Per Karsson

**Affiliations:** 1Department of Oncology, Institution of Clinical Sciences, Sahlgrenska Academy, Sahlgrenska University Hospital, University of Gothenburg, 413 45 Gothenburg, Sweden; anna.ofverholm@vgregion.se (A.Ö.); zakaria.einbeigi@oncology.gu.se (Z.E.); erik.holmberg@gu.se (E.H.); 2Faculty of Medicine, Umeå University, 901 87 Umeå, Sweden; antonia.wigermo@gmail.com

**Keywords:** BRCA1, BRCA2, hereditary breast cancer, hereditary ovarian cancer, risk reducing mastectomy, risk reducing salpingo-oophorectomy, survival

## Abstract

Women with *BRCA* variants have a high lifetime risk of developing breast and ovarian cancer. The aim of this study was to investigate the standard incidence ratios (SIR) for breast and ovarian cancer and standard mortality ratios (SMR) in a population-based cohort of women in Western Sweden, under surveillance and after risk reducing surgery. Women who tested positive for a *BRCA* variant between 1995–2016 (*n* = 489) were prospectively registered and followed up for cancer incidence, risk reducing surgery and mortality. The Swedish Cancer Register was used to compare breast and ovarian cancer incidence and mortality with and without risk reducing surgery for women with *BRCA* variants in comparison to women in the general population. SIR for breast cancer under surveillance until risk-reducing mastectomy (RRM) was 14.0 (95% CI 9.42–20.7) and decreased to 1.93 (95% CI 0.48–7.7) after RRM. The SIR for ovarian cancer was 124.6 (95% CI 59.4–261.3) under surveillance until risk reducing salpingo-oophorectomy (RRSO) and decreased to 13.5 (95% CI 4.34–41.8) after RRSO. The SMR under surveillance before any risk reducing surgery was 5.56 (95% 2.09–14.8) and after both RRM and RRSO 4.32 (95% CI 1.62–11.5). Women with cancer diagnoses from the pathology report after risk reducing surgery were excluded from the analyses. Risk reducing surgery reduced the incidence of breast and ovarian cancer in women with *BRCA* variants. However, overall mortality was significantly increased in comparison to the women in the general population and remained elevated even after risk reducing surgery. These findings warrant further research regarding additional measures for these women.

## 1. Introduction

The *BRCA1* and *BRCA2* genes were discovered in the early 1990s [1,2]. In Sweden, oncogenetic clinics were established at university hospitals in the six national healthcare regions (North, Uppsala-Örebro, Stockholm, West, Southeast, and South) as *BRCA* screening was implemented for women with hereditary breast and ovarian cancer. Subsequently, each oncogenetic clinic was comprised of a team of oncologists, clinical geneticists, clinical laboratory geneticists and oncology nurses that could provide genetic counseling and screening for patients and families with hereditary breast and ovarian cancer. The oncogenetic clinic at Sahlgrenska University Hospital in Gothenburg, Sweden was established in 1995 and currently serves Western Sweden with a population of 1.9 million inhabitants. Intriguingly, a novel recurrent founder variant in *BRCA1* with origins in Western Sweden was discovered during the first years of *BRCA* screening at the Clinical Genetics laboratory [3,4]. During the first years of BRCA-testing, about 77% of women positive for any disease-causing variant in Western Sweden carried this variant [4]. A regional follow-up register was initiated for women with *BRCA* variants at the oncogenetic clinic to ensure that these women were given access to relevant clinical surveillance programs, and to be able to evaluate the long-term effects of measures, e.g. risk reducing surgery, performed within the cohort. Referral criteria for the assessment of hereditary breast and ovarian cancer has followed previous and current national healthcare programs (https://www.cancercentrum.se), and are in general similar to international guidelines (https://www.nccn.org).

In a prospective cohort study with data from three international consortia, the cumulative risk for breast and ovarian cancer at 80 years of age for women with *BRCA* variants were estimated. The risks for *BRCA1*-carriers were 72% (95% CI, 65–79%) and 44% (95% CI, 36–53%), and for *BRCA2*-carriers 69% (95% CI, 61–77%) and 17% (95% CI, 11–25%), respectively [5]. Risk reducing mastectomy (RRM) and risk reducing salpingo-oophorectomy (RRSO) substantially reduced these risks. A systematic review of risk reducing surgery in women with *BRCA* showed a risk reduction of 90–100% of breast cancer after RRM and 72–88% for ovarian cancer [6]. One study showed an annual risk of peritoneal cancer after RRSO of 0.20% for women with *BRCA1* and 0.10% for women with *BRCA2*, and in another study there was a 20-year cumulative risk of peritoneal cancer of 3.5% [7,8].

Further, RRSO substantially reduced the increased mortality, both breast cancer–specific and ovarian cancer–specific mortality and all-cause mortality [6,8,9]. To our knowledge, there are however no reports in the literature showing cancer incidence and mortality among women with *BRCA* variants after risk reducing surgery in a defined geographic region in comparison with the general female population in the same region.

National guidelines regarding hereditary breast and ovarian cancer recommends annual diagnostic imaging with breast magnetic resonance imaging (MRI) and breast ultrasound from ages 25 to 55 and mammography from ages 25 to 74 years. RRM has routinely been offered as a choice for those women who wanted a higher degree of risk reduction instead of surveillance. Regarding the ovaries, annual gynecological ultrasound was offered from 30 years of age. But more importantly, RRSO was recommended from 35 years of age for women with *BRCA1* variants and from 40 years of age for women with *BRCA2* variants [8,10]. During the surveillance period of this study the standard surgical procedure for RRM was a complete removal of breast tissue, initially including removal of the nipple but later changing to nipple sparing technique. The standard surgical procedure for RRSO has been baging and removal of the ovaries and fallopian tubes with laparoscopic techniques.

The aim of the present study was to assess the impact of risk reducing surgery on cancer incidence and mortality during a 20-year period (1995–2016) in a geographically defined population-based cohort of women in Western Sweden with *BRCA* variants. We evaluated cancer incidence and mortality among cancer-free women who underwent RRM and RRSO in comparison with baseline risks without risk reducing surgery and in comparison with the general female population.

## 2. Materials and Methods

The study population consisted of women that had tested positive for disease causing variants in *BRCA* between 1995–2016 at the regional oncogenetic clinic in Western Sweden and were given information about participating in a follow-up register to evaluate the long-term effects of surveillance and risk reducing surgery. In total, 489 women with *BRCA* variants were included in the register. The register contained information on date of DNA analysis, the identified *BRCA* variant, date of any previous malignancy, date of annual breast MRIs, ultrasounds, mammograms, date of annual gyneacological ultrasounds, date of risk reducing mastectomy, date of risk reducing oophorectomy, date and type of any cancer diagnosis after registration, date of death, and cause of death. Approval of the register was obtained from the Swedish Central Ethical Review Board (registration number EPN S331-01 2001, EPN S331-01 2017), and all patients included in the register signed a consent form.

Patient clinical data in the register was curated during June–July 2017, compared to medical files, and was supplemented when required. The cohort consisted of the 489 women who were registered as positive for disease causing *BRCA* variants during this 20-year period, either via *BRCA* screening or tested for a known familial variant. The analytical cohort consisted of the 253 women who were cancer-free when registered (Figure 1).

In total, 179 of the 489 women had been diagnosed with breast cancer only before the DNA analysis was performed, 35 had ovarian cancer only and 22 had both breast and ovarian cancer before DNA analysis, leaving 253 cancer-free women at the time of DNA analysis. Consequently, the 253 patients formed the analytic cohort, and consisted of 213 *BRCA1*- and 40 *BRCA2*-carriers.

The median age for DNA analysis in this cohort was 36.8 years of age (range, 20.3–78.9). The median age at the end of follow-up was 46.9 years of age (range, 22.2–87.3). All instances of breast and ovarian cancer diagnosed after the DNA analysis, i.e., during the surveillance period, were classified as diagnosed before, at, or after risk reducing surgery. A diagnosis at risk reducing surgery was defined as diagnosed from the final pathology report or within 90 days after surgery. These cases were excluded from the analysis of SIR for breast and ovarian cancer and for the analyses of SMR.

Four of the 253 women had already undergone a bilateral RRM before the time of DNA analysis, and therefore 249 women and 498 breasts were under surveillance after the time of DNA analysis. Fourteen of the 253 women had undergone a RRSO before the time of DNA analysis, and therefore 239 women with ovaries were under surveillance after the time of DNA analysis. When analyzing breast cancer incidence for women with breasts, breast years were calculated from the time of DNA analysis to the first of following events; diagnosis of breast cancer, RRM, death or end of surveillance period, i.e., 31 December 2016. Every breast was accounted for separately. For women who underwent RRM, breast years were calculated from the time of RRM to the first of following events; diagnosis of breast cancer, death or end of surveillance period, i.e., 31 December 2016. The SIR was defined as the observed number of breast cancers during these breast years divided by the expected number of cases, using incidence rates from the Swedish female population stratified for 5-year age groups (0–4, 5–9, … 80–84, 85–) and calendar year.

When analyzing ovarian cancer incidence for women with their ovaries intact, person years were calculated from the time of DNA analysis to the first event; ovarian cancer, RRSO, death or end of surveillance period. For women who underwent RRSO, person years were calculated from the time of RRSO to the first event; peritoneal cancer assessed to be of ovarian origin, death or end of surveillance period. SIR was defined as the observed number of peritoneal cancers of ovarian origin during these person years divided by the expected number of cases, using incidence rates from the Swedish female population stratified for 5-year age groups (0–4, 5–9, … 80–84, 85–) and calendar period. Incidence rates for breast and ovarian cancer produced by the NORDCAN project [11] were used. The coding of cancer in our cohort was the same as used by the NORDCAN project and followed definitions according to International rules for multiple primary cancers [12].

Events of peritoneal cancer of unknown origin were not included in the analyses of SIR and SMR but were described in the results section. The reason for this was that the incidence rate of peritoneal cancer in the background population is uncertain.

When analyzing the mortality rate for all women in the cohort, person years were calculated from the time of DNA analysis to death or end of surveillance period. When analyzing the mortality rate for women before any risk reducing surgery, person years were calculated from the time of DNA analysis to bilateral RRM or RRSO or death or end of surveillance period. When analyzing the mortality rate for women after bilateral RRM or after RRSO, person years were calculated from bilateral RRM or RRSO to death or end of surveillance period. Lastly, when analyzing the mortality rate for women after both bilateral RRM and RRSO, person years were calculated from the last risk reducing surgery event to death or end of surveillance period. The SMR was defined as the observed number of deaths during these person years divided by the expected number of deaths, using rates from the Swedish female population stratified for 5-year age groups (0–4, 5–9, …, 80–84, 85–) and calendar year.

The stset, stsplit and strate macros in STATA 15.1 were used for the incidence analyses [13].

## 3. Results

### 3.1. Breast Cancer Incidence

In total, 31 breast cancers were diagnosed for 249 women who were originally cancer-free when registered, of which 25/31 were diagnosed during surveillance, 4/31 were diagnosed in the final pathology report from RRM and 2/31 cancers were diagnosed after the women underwent a RRM (Table 1).

The median age at diagnosis was 46.0 years (range, 32.4–64.4) and the median age for RRM was 42.3 years (range, 22.0–65.3). Thirty women had 31 breast cancer diagnoses; 27 cancers were associated to *BRCA1*, 4 were associated to *BRCA2*. Also, 25 breast cancers were observed in women under surveillance, while 1.79 were the number expected, resulting in a SIR of 14.0 (95% CI, 9.46–20.7; Table 2). The three youngest women out of the 25 cases were diagnosed between 30 and 34 years of age.

Eighty women had a bilateral RRM during surveillance, whereas 24 women had a unilateral RRM after being diagnosed with breast cancer during surveillance, i.e., in total 184 breasts. Two cancers were observed after risk reducing surgery while the expected number was 1.0, SIR 1.93 (95% 0.48–7.70). The youngest woman was diagnosed at age interval 35–39, and one woman at age interval 45–49.

### 3.2. Ovarian Cancer Incidence

Fourteen of the 239 women were diagnosed with ovarian cancer, or peritoneal cancer of ovarian origin. Seven of the 14 cancers were diagnosed during surveillance, 4/14 cancers were findings from the final pathology report from RRSO, and 3/14 cancers were diagnosed after the women had RRSO (Table 1). In addition to these 3 cases, there were 3 other cases of peritoneal cancer with unknown origin. These 3 other cases were not included in the analyses. The median age at diagnosis was 46.7 years (range, 39.1–51.5) and the median age at RRSO was 43.4 years (range, 28.2–79.7). All 14 women with ovarian cancer had *BRCA1* variants. Seven of the 14 ovarian cancers were observed during surveillance, while the expected number was 0.06, with a SIR of 124.57 (95% CI, 59.39–261.30; Table 2). The youngest woman was diagnosed in the 35–39 age interval. Among the 136 women who had undergone a RRSO, three cancers were observed during follow-up after the RRSO, while the expected number was 0.22, resulting in a SIR of 13.47 (95% CI, 4.34–41.76). All women were diagnosed at the 40–49 age interval.

### 3.3. Overall Mortality

Of the 253 cancer-free women at time of DNA analysis, 16 died during the 20-year surveillance period (Table 3). Three women died from breast cancer, 1 of them from bilateral breast cancer. Seven women died from ovarian cancer, including 5/7 from peritoneal cancer of ovarian origin after having undergone a RRSO before or after the time of DNA analysis. Two women died from lung cancer and 1 died from cervical cancer, while 3 died from other causes than malignancies (intracerebral bleeding, cardiovascular disease and chronic obstructive pulmonary disease, dementia and ileus). In total, 13 women had a *BRCA1* variants, while 3 had a *BRCA2* variant.

Analysis of overall mortality showed that 16 women died during the observation period, while the expected number of deaths was 4.9, SMR 3.24 (95% CI 1.99–5.30; Table 2). The 2 youngest women who died were in the 40–44 age interval. Of the 235 women without any previous risk reducing surgery at the time of DNA analysis, 4 died during surveillance, while the expected number was 0.72, SMR 5.56 (95% CI 2.09–14.8). The youngest woman who died was in the 45–49 age interval.

Of the 80 women who had a bilateral RRM, 4 deaths were observed during follow-up, while the expected number of deaths was 1.0, SMR 4.15 (95% 1.56–11.0). Also, 3/4 of the deceased women were in the 45–49 age interval, and breast cancer was the cause of death for only one of them. In total, 136 women had a RRSO and during follow-up 11 deaths were observed, while the expected number was 3.7, SMR 2.99 (95% CI 1.66–5.40). The youngest patient to die was in the 40–44 age interval and four in the 45–49 age interval. Five of the 11 deaths after RRSO were due to ovarian or peritoneal cancer.

Among 62 women that had undergone both RRM and RRSO, 4 deaths occurred, 3 from peritoneal cancer of ovarian origin and 1 with a non-malignant cause of death, SMR 4.32 (95% CI 1.62–11.5). Three of 62 women had an ovarian cancer diagnosis from the final pathology report and as previously mentioned, these were excluded from the analyses of SIR and SMR. These women did not die during the surveillance period.

## 4. Discussion

This population-based follow-up study was restricted to a defined geographic region, Western Sweden, with a cohort of women with *BRCA* variants, primarily registered at the time of genetic testing and thereafter prospectively followed regarding cancer incidence, risk reducing surgery and death between 1995–2016. In concordance with previous studies, we observed a reduction in breast and ovarian cancer incidence after RRM and RRSO as well as mortality [6,8,9]. However, the overall mortality rates for women with *BRCA* variants were still significantly increased compared to age matched-women in the general population, even after risk reducing surgery, SMR 4.32 (95% CI, 1.62–11.5). As far as we know, a persistent increase in SMR even after risk reducing surgery has not been previously shown in a study with a population-based cohort from a geographically defined region. These findings emphasize the need for further research to identify additional complementary measures than risk reducing surgery for high-risk women with *BRCA* variants.

For example, a more precise and individualized risk evaluation combining genetic screening for polygenic risk score in combination with hormonal factors and family pedigree could influence the woman to take preventive measures; engaging in surveillance programs or having risk reducing surgery, in time [14]. Further, ovarian cancer may arise in the fallopian tubes [15] which raises the question if sequential removal of tubes and ovaries would decrease their risk of ovarian cancer. PARP-inhibition and antihormonal drugs may with the right timing help in preventing cancer incidence and mortality [16]. RRSO for premenopausal women results in an immediate onset of menopause, which can bring the physical and emotional symptoms of natural menopause, and there is also an elevated risk of osteoporosis and probably also cardiovascular disease [17]. Hormone replacement therapy can be offered to women after RRSO to partly counteract the symptoms. Still, the negative aspects of early-induced menopause, including the impact on quality of life, are not fully known [18] and further preventive studies among these patients is warranted. Furthermore, early detection of cancer as well as early detection of a relapse of treated cancer through biomarkers such as circulating tumor cells, cell free tumor DNA and micro-RNA in blood may improve the possibilities for earlier intervention and treatment [19,20].

In our cohort, we observed two cases of breast cancer among 80 women who had a bilateral RRM. Furthermore, we observed three cases of peritoneal cancers of ovarian origin after RRSO among 136 women. This is in line with studies showing no or small remaining risks for cancer [6,7] after risk reducing surgery. Patients as young as 30–34 and 35–39 years of age at the time of diagnosis were identified with breast and ovarian cancer, respectively. Current clinical guidelines recommend starting surveillance of the breasts at age 25 and for the ovaries at age 30, which was in concordance with the findings in this study. There is no age recommendation for RRM, but a firm recommendation to consider RRSO from 35 years of age for *BRCA1* and 40 years of age for *BRCA2*. These recommendations were thus adequate in relation to the ages of cancer diagnoses in our cohort.

In this study cohort of 253 women, 213 had *BRCA1* variants and only 40 had *BRCA2* variants, which affects the generalizability as *BRCA1* is associated with a higher risk of both breast and ovarian cancer. Further, there is in the literature evidence for variant genotype–phenotype correlations in *BRCA* genes [21]. Four out of 10 women with *BRCA1* variants carried the founder variant c.3171ins5 with an origin in Western Sweden, but if there is a correlation between this very variant and the breast and ovarian cancer ratios, is not known. *BRCA* variants may be associated with other cancers such as pancreas and colon cancer, in particular *BRCA2* variants are associated to a relatively low increase in risk of pancreas cancer [22]. Those two cancer diagnoses were absent in our analytical cohort.

All women in the region of Western Sweden with a *BRCA* variant were asked to participate in a prospective follow-up register study. There were only a few lacking consents and the genetic counselors did not perceive the lacking consents to be related to whether the patient had cancer or risk reducing surgery. The high degree of accepting registration and surveillance in our cohort may depend on the fact that healthcare in Sweden is mainly financed by taxes, access to healthcare is subsidized or free for the patient, and the healthcare region is responsible for funding and providing healthcare services, including this type of surveillance program, risk reducing surgery and cancer treatment.

A strength of this study was the prospectively-followed primarily registered cohort of 489 women with *BRCA* variants in a defined geographic region during the period 1995–2016. Follow up was complete and the medical exams during surveillance as well as risk reducing surgery were well described in the register. Limitations of the study was the limited number of person years of follow-up and limited time of follow-up due to the majority of the women entering the study during the latter part of the inclusion period, at which time genetic testing gradually became more common. This resulted in wide confidence intervals for the estimates.

## 5. Conclusions

In this population-based study, women with *BRCA* variants had a remaining increased mortality in comparison to women in the general population, even after risk reducing surgery. This persistent increase in mortality calls for further measures than just risk reducing surgery in the risk management of women with *BRCA* variants.

## Figures and Tables

**Figure 1 genes-10-01046-f001:**
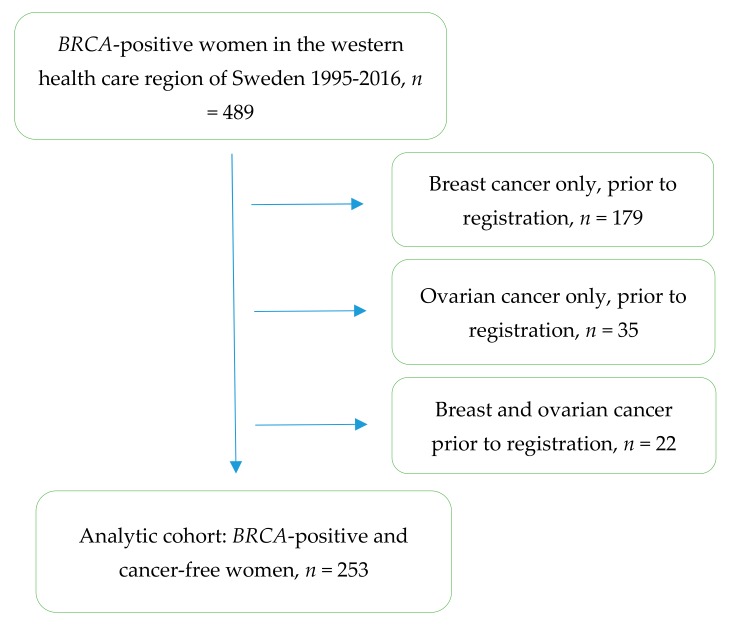
All *BRCA*-positive women registered in western Sweden from 1995–2016 (*n* = 489). The analytic cohort consists of all those *BRCA*-positive women who were cancer-free when registered (*n* = 253).

**Table 1 genes-10-01046-t001:** Diagnoses of breast and ovarian cancer, including peritoneal cancer of ovarian origin, among cancer-free women under surveillance.

**Breast cancer in the cohort**	
All women, cancer-free when registered	253
All women cancer-free and with both breasts	249
Breast cancer diagnoses	31/249
Diagnoses during surveillance	25/31
Diagnoses from final pathology report, RRM	4/31
Diagnoses after RRM	2/31
Mean age at diagnosis (range)	46.0 (32.4–64.4)
Median age at RRM	42.3 (22.0–65.3)
**Ovarian cancer in the cohort**	
All women, with ovaries and cancer-free when registered	239
Ovarian cancer diagnosis	14/239
Diagnosis during surveillance	7/14
Diagnosis from final pathology report, RRSO	4/14
Diagnosis after RRSO	3/14
Mean age at diagnosis (range)	46.7 (39.1–51.5)
Median age at RRSO	43.4 (28.2–79.7)

**Table 2 genes-10-01046-t002:** Women with *BRCA* variants, cancer-free when registered, during surveillance, before and after risk reducing mastectomy (RRM) and/or risk reducing salpingo-ophorectomy (RRSO): Standard incidence ratio (SIR) of breast and ovarian cancer, including peritoneal cancer assessed to be of ovarian origin after RRSO, and standard mortality ratio (SMR).

**Breast Cancer Incidence**	**Follow-Up Time, Years, Median (Range)**	**Breast Years**	**Number of Breast Cancers (Expected Number)**	**SIR (95% CI)**
Women before RRM (*n* = 249)	3.85 (0.02–19.9)	2721.6	25 (1.8)	14.0 (9.42–20.7)
Women after RRM (*n* = 184)	5.3 (0.3–23.8)	1271.7	2 (1.04)	1.93 (0.48–7.70)
**Ovarian Cancer Incidence**	**Follow-Up Time, Years, Median (Range)**	**Person Years**	**Number of Ovarian Cancers (Expected Number)**	**SIR (95% CI)**
Women before RRSO (*n* = 239)	2.2 (0.0–17.3)	887.4	7 (0.1)	124.6 (59.4–261.3)
Women after RRSO (*n* = 136)	7.6 (0.58–23.8)	1883.9	3 (0.2)	13.5 (4.34–41.8)
**Mortality**	**Follow-Up Time, Tears, Median (Range)**	**Person Years**	**Number of Deaths (Expected Number)**	**SMR (95% CI)**
All women in cohort (*n* = 253)	7.6 (0.8–21.1)	2160.6	16 (4.9)	3.24 (1.99–5.30)
Women before any risk reducing surgery (*n* = 235)	2.1 (0.0–17.7)	833.6	4 (0.7)	5.56 (2.09–14.8)
Women after RRM (*n* = 80)	5.3 (0.6–19.1)	577.6	4 (1.0)	4.15 (1.56–11.0)
Women after RRSO (*n* = 136)	7.4 (0.7–21.1)	1144.3	11 (3.7)	2.99 (1.66–5.40)
Women after both RRM and RRSO (*n* = 62)	6.5 (0.9–19.3)	480.5	4 (0.9)	4.32 (1.62–11.5)

**Table 3 genes-10-01046-t003:** Causes of death for the 16 of the 253 women who were cancer-free when registered and died during surveillance.

Cause of Death in the Cohort	
Breast cancer	3/16
Breast cancer after RRM	0/3
Ovarian cancer	7/16
Peritoneal cancer of ovarian origin after RRSO	5/7
Other cancer	3/16
Other cause, non-malignant	3/16

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
