# Peer review of "Increased Overall Mortality Even after Risk Reducing Surgery for BRCA-Positive Women in Western Sweden"

_genes, 2019, doi:10.3390/genes10121046_

Round 1

Reviewer 1 Report

The authors provide data on an apparent lack of benefit on mortality of having risk reducing surgery in a relatively small number of BRCA1/2 female carriers unaffected at the time of testing. Whilst this is a potentially important paper it is vital that the authors clearly separate out those who have been diagnosed with cancer at either risk reducing operation. 67% of the ‘prospective’ breast cancers were prevalent at RRM with only two prospective. Similarly 4/7 ‘ovarian cancer’ were diagnosed at RRSO. By including these ‘prevalent’ cancers in the calculations the authors are not truly assessing the benefits POST surgery. It is of course relevant to overall benefit of RRS however including the prevalent cancers in calculating mortality will seriously underestimate the longer term benefits once the prevalence effect is removed. For instance women can be told that if they are cancer free after RRM/RRSO their mortality is reduced to xx. It is clear that a press release on the current abstract would provide a very misleading picture to women who have had both surgeries and are cancer free.

Specific points

‘For women who underwent RRSO, person years were calculated from the time Breast of RRSO to the first event (ovarian) cancer, death or end of surveillance period’- suggest replace ‘ovarian’ with ‘primary peritoneal’. ‘Lastly, when analyzing the mortality rate for women after both bilateral RRM and RRSO, person years were calculated from the last risk reducing surgery event to death or end of surveillance period’ –This should explicitly EXCLUDE any who have developed cancer at first risk reducing surgery or between surgeries. Please clarify that all risk reducing surgeries included the fallopian tubes For cancer incidence after surgery ‘prevalent’ cancers at surgery should be excluded. Please make this clear in methods. It is correct in tables and results The authors must make it clear how many of the five with true post Risk reducing surgery have died from cancer and how many of the prevalent cancers. A separate calculation is required for those who did not have ‘prevalent’ cancers at surgery and this should be in the abstract. I am very concerned that 3 peritoneal cancers have occurred after RRSO in only 136 women. We have had zero such cancers in 820 BRCA carriers with 7 times the follow up. This raises substantial doubts about the quality of the RRSO. Much of the only 80% reduction is based on older papers where the need to bag the ovaries and perform peritoneal lavage was not preformed. ‘Studies comparing the risks between women with BRCA variants having undergone risk reducing surgery with those just on surveillance indicated lower breast cancer-specific mortality, ovarian cancer-specific mortality and overall mortality (8, 9).’ –This was true for reference 9 only for BRCA1. The very low number of BRCA2 carriers and the very specific nature of a single BRCA1 PV need to be added as a further limitation to generalisability.

Reviewer 2 Report

This is a very interesting epidemiology study in which the Authors showed increase mortality ratio in BRCA positive patients with or without risk reducing surgery compared to the general population in The Swedish Cancer Register.
This finding is very important considering the still controversial management of these Patients and the unknown survival impact of the risk reducing procedures.

There are some points that the Authors should better address/comment in this study
Line 54-55
“ reduced these 54 risks (6, 7) with more than 90% for breast cancer (8) and with 80% for ovarian or for peritoneal 55 carcinomatosis”
These percentages seemed to be old and not coherent with modern literature. The modern risk reducing surgery seems to have less risk for ovarian and breast cancer.

An important comment in the discussion would need to address the need of appropriate risk reducing surgery technique to avoid possible cancer in spite the risk reducing surgery.

Another important data not clear in this study is the age the BRCA population had the risk reducing surgery, accounting best benefit (longest gain in life years) sooner in the age the surgery was done.

Also, very important is a comment of gain in quality of life years, that probably cannot be addressed by these data, but may deserve a comment into the discussion.

We know that BRCA women may have increase incidence of other cancers with poor prognosis (pancreatic, colon). Not clear from the paper if this was the case in the BRCA patient who underwent risk reducing surgery. We may comment that mortality for other cancers may overcome the possible mortality for breast and ovarian cancer.

Another not clear aspect in the paper is the concept of year gained after the risk reducing surgery.
Was mortality of the risk reducing surgery woman later in their age because the procedure, or they died anyway younger with respect to non-operated or general population woman.

Line 236-238
Furthermore, early detection of cancer 236 as well as early detection of a relapse of treated cancer through biomarkers such as circulating tumor 237 cells, cell free tumor DNA and micro-RNA in blood may improve the possibilities for earlier 238 intervention and treatment (20, 21).

This may apply also to the general population, but we do not have any evidence that this would happen soon.

Round 2

Reviewer 1 Report

The authors have clarified a number of issues in relation to studty design and that the cancers at surgery were excluded from analysis. I am still very conncerned at the high rate of peritoneal cancers after surgery which is unusual and is the major cause of a loss of a survival benefit.

A further comment that their peritoneal primary rate despite bagging was high is justified baging needs to be changed to 'bagging' Gene symbols in the new text need italicising

Reviewer 2 Report

The Authors were able to address all the points of the reviewers